# Bistable and photoswitchable states of matter

Brady T. Worrell[1], Matthew K. McBride[1], Gayla B. Lyon[1,7], Lewis M. Cox[1,2,3], Chen Wang[1,7], Sudheendran Mavila[1], Chern-Hooi Lim[1,8], Hannah M. Coley[4], Charles B. Musgrave[1], Yifu Ding[3,4] & Christopher N. Bowman [1,4,5,6]

Classical materials readily switch phases (solid to fluid or fluid to gas) upon changes in pressure or heat; however, subsequent reversion of the stimulus returns the material to their original phase. Covalently cross-linked polymer networks, which are solids that do not flow when strained, do not change phase even upon changes in temperature and pressure. However, upon the addition of dynamic cross-links, they become stimuli responsive, capable of switching phase from solid to fluid, but quickly returning to the solid state once the stimulus is removed. Reported here is the first material capable of a bistable switching of phase. A permanent solid to fluid transition or vice versa is demonstrated at room temperature, with inherent, spatiotemporal control over this switch in either direction triggered by exposure to light.

[1] Department of Chemical and Biological Engineering, University of Colorado, Boulder, CO 80309, USA. [2] Applied Chemicals and Materials Division, National Institute of Standards and Technology, Boulder, CO 80305, USA. [3] Department of Mechanical Engineering, University of Colorado, Boulder, CO 80309, USA. [4] Material Science and Engineering Program, University of Colorado, Boulder, CO 80309, USA. [5] BioFrontiers Institute, University of Colorado, Boulder, CO 80309, USA. [6] Department of Restorative Dentistry, University of Colorado, Anschutz Medical Campus, Aurora, CO 80045, USA. [7] Present address: Formlabs Inc, 35 Medford St. #201, Somerville, MA 02143, USA. [8] Present address: Department of Chemistry, Colorado State University, Fort Collins, CO 80523, USA. Correspondence and requests for materials should be addressed to C.N.B. (email: Christopher.Bowman@colorado.edu)

Matter readily switches phases (solid to fluid or fluid to gas) upon changes in pressure or heat; however, subsequent reversion of the stimulus returns the material to its original phase.[1] Covalently cross-linked polymer networks (i.e., thermosets), which are solids that do not irreversibly flow or deform when a stress is applied, do not switch phase even with changes in temperature and pressure.[2] Conversely, upon addition of covalent cross-links with a dynamic character into the polymer network they, much like classical materials, switch from a solid to a fluid when the dynamic crosslinks are activated; however, due to the types of dynamic chemistry that have hitherto been explored, removal of the stimulus inevitably returns the material to the solid state (Fig. 1a).[3–15]

Covalently cross-linked polymers capable of switching phases (solid to fluid and vice versa), a set of materials often referred to as covalent adaptable networks (CANs),[15] are, as stated, controllable by the application of an external stimulus such as heat or light that activates the dynamic covalent chemistry (DCC) within the network, temporarily switching the material from solid to fluid (Fig. 1a). In the first instance where heat acts as the stimulus, techniques developed by Leibler,[3–5] Du Prez[6,7] and others have utilized degenerate exchange reactions with high kinetic barriers; heating these so-called vitrimers[3–7] to sufficiently high temperatures overcomes these barriers and facilitates rapid bond exchange, effectively fluidizing the material, while cooling returns the network to a solid state. Approaches pioneered by Bowman,[8,9] Matyjaszewski,[10–12] and others have used light to rapidly produce thiyl or carbon centered radicals capable of adding to unsaturated species within the cross-link to cause statistical fragmentation, temporarily forming a viscoelastic fluid. Turning off the light results in rapid termination of radicals, thus eliminating the addition-fragmentation bond exchange sequence which returns the material to an elastic solid state. In either instance switching of the phase, from a solid to a fluid, only occurs while the stimulus is being applied to the material as high energy intermediates are required to be formed for fluidic behavior to persist. Thus, a platform which enables thermosetting polymers, without any other changes to their chemical structure, to switch phase permanently and in a bistable manner following a short, transient exposure to an external stimulus has not been developed or explored.

Here, we demonstrate responsive materials based on similar chemistry with switches phases, solid to fluid or vice versa, at room temperature upon exposure to light; this phase change is bistable, permanent, and removal of the stimulus does not return the materials to their previous state (Figs. 1b, c).[16] This room temperature change in phase is feasible due to the incorporation of a thioester functional group placed throughout the polymer network which engages in a remarkably robust dynamic exchange reaction with free thiol as promoted by a base catalyst.[17–19] This dynamic exchange was found to be promoted or halted by mild basic or acidic catalysts, respectively, which are released by light.[20,21] Furthermore, the photo-mediated release of these catalysts has inherent spatial and temporal control over the phase of the material, allowing for a singular material to have a controlled multitude of volumes which act as fluids or solids in a bistable manner as directed by light (Fig. 1c). Our results demonstrate that light can be employed to permanently transition where and when this switchable material acts as either a fluid or a solid (Fig. 1b).

## Results

### Development of a photoswitchable material.
Considering these previously reported systems, we postulated that a bistable material capable of permanent, photoinduced switching of phase could

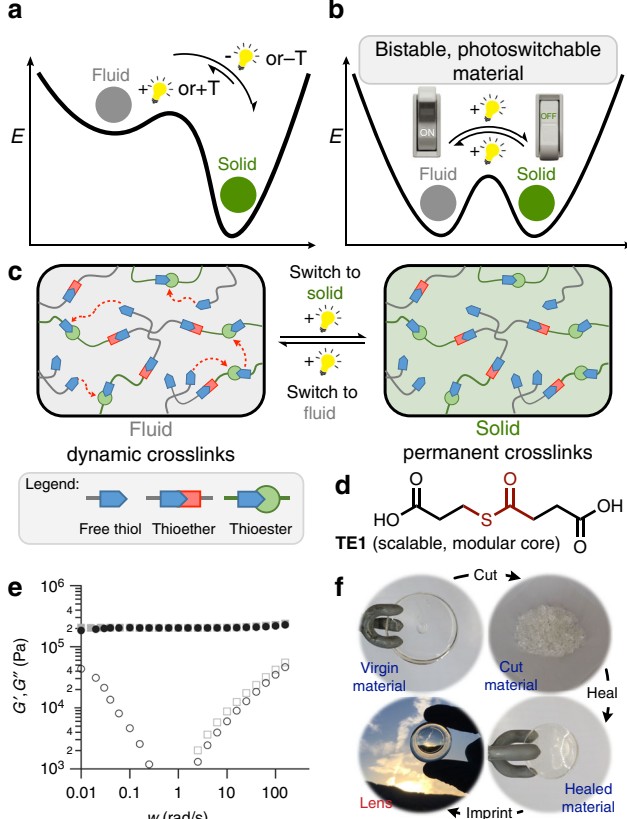

**Fig. 1** Concept of a bistable, photoswitchable state of matter. **a** The application of a given stimulus (i.e., heat or light) to a solid material overcomes the unfavorable solid to fluid phase transition in covalent adaptable networks; removal of the stimulus returns the material to the favored solid phase. **b** Equivalent energetic favorability of the fluid and solid phase of a covalent adaptable network with a small barrier allows for permanent, bistable switching of phase with light. **c** A cartoon of how light can effectively switch the state of matter in a network polymer. **d** Rheometry shows that thioester crosslinked materials act as fluids when thiol, thioester, and a base catalyst (PMDETA) are present (G' = black filled circle; G" = black open circle), and as solids when the base catalyst is removed (G' = grey filled square; G" = grey open square). **e** The ability of thioester crosslinked materials while in the fluid phase to undergo large changes in structure at room temperature is indicated by the ability to shred and subsequently heal the material into a defect free, optically active material

be achieved if the strengths of both thermal and light-induced bond exchange approaches were combined. Namely, we sought to develop a degenerate anionic exchange reaction with a low kinetic barrier (active at room temperature, ~23 °C) that requires a persistent catalyst which is readily generated or consumed using light without altering the structure of the network. Here, we have developed a unique approach using a thioester containing monomer which is seamlessly incorporated into standard thiol-X[22] and other polymerizations to rapidly form cross-linked network polymers where a residual amount of thiol remains unreacted. Such networks were shown to exhibit rapid ambient temperature fluidic properties in the presence of a base catalyst via the thiol-thioester exchange (Supplementary Figure 7, the mechanism of the thiol-thioester exchange).[17–19] Within these networks, the exchange was found to be chemoselective and proceed rapidly at room temperature. As the exchange is promoted with very mild catalysts, light was employed to temporally

and instantaneously provide or deprive the material of these reagents. Spatial fluidization (turning ON exchange) or solidification (turning OFF exchange) was further evidenced in macroscopic and microscopic demonstrations towards applications in photonics, smart coatings, and bulk materials.

To form suitable thiol-thioester exchange-based polymer networks, monomers had to be synthesized which (i) contained a thioester that was active in the thiol-thioester exchange, (ii) were decorated with functionality capable of engaging in thiol-X (or other suitable) polymerization chemistry, (iii) were highly scalable, and (iv) were bench-stable. After several iterations towards realizing a monomer which fit each of these requirements, the scalable synthesis of the core monomer **TE1** was devised. This simple thioester containing molecule is flanked by two readily transformable carboxylic acids and was utilized as a staple molecule in all subsequent studies. Moving towards a monomer compatible with thiol-X polymerizations,[22] thioester **TE1** was found to undergo facile Fisher esterification with allyl alcohol to deliver **TE2**, a thioester flanked by two allyl esters, in excellent yield. Accordingly, combination of our diene monomer **TE2** (1 equiv) with a commercial tetrafunctional thiol (pentaerythritol tetra(3-mercaptopropionate or PETMP, 1 equiv. w/r/t monomer), a weakly basic amine (N,N,N′,N″,N″-pentamethyldiethylenetriamine or PMDETA, 20 mol%), and a UV-photoinitiator (DMPA, 1 mol%) gave a homogeneous, non-viscous resin which could readily be cast into a thin film or mold. Upon irradiation of this resin, a rapid, quantitative thiol-ene reaction took place (365 nm, 50 mW/cm$^2$, <5 s irradiation, Supplementary Figure 9, FT-IR kinetics of the reaction),[23] leaving a persistent amount of thiol unreacted in the network to serve in the thiol-thioester exchange (Supplementary Figure 10, overlaid kinetics and conversion for each formulation). As shown using rheological experiments, networks which contained thioester cross-links, free thiol, and base catalyst acted as a viscous fluid at room temperature under low frequency shearing (~1 rad//, Fig. 1c). This increase in loss modulus (G″) is directly related to bond exchange occurring in the network, behaving as a viscous fluid and not a viscoelastic solid. In contrast, removal of the base catalyst resulted in networks which behaved as typical cross-linked solids, with the storage modulus (G′) staying constant and the loss modulus (G″) decreasing at lower frequencies (Fig. 1c). Creep experiments further illustrated that only networks which contained all components demonstrated fluidic properties at room temperature; indeed, the creep behavior resembled that of an entangled polymer melt with a linear increase in strain with time and a creep rate of 0.59%/min (at 42 kPa, Supplementary Figure 11). In this experiment the thioester control (– thioester) was a diallyl ester of a similar length as **TE2** where the thioester functional group was replaced by two methylene (CH$_2$) groups (**DAEC**, Supplementary Figure 11). Increasing the base (PMDETA) concentration resulted in more rapid network rearrangement, presumably due to the proportionally higher concentration of thiolate present (Supplementary Figure 12, overlaid stress relaxation graph). Covalent attachment of the base (PMDETA, via a thiol-ene reaction with a pendant ene) to the network did not affect the ability of the network to relax stress (Supplementary Figure 13, overlaid stress relaxation graph). Heating the network was found to increase the stress relaxation rate, presumably due to increased mobility of the chain ends at higher temperatures (Supplementary Figure 14, overlaid stress relaxations ran at 25 °C, 50 °C, and 75 °C). To demonstrate the ability of these fluid-like materials to undergo large changes in structure, a disk of material was polymerized, cut into small pieces, remolded at room temperature, and imprinted into a lens. Significantly, the material was able to heal to a defect free polymer network with no change in the material's optical clarity being observed (Fig. 1d).

Although weak bases, such as PMDETA, were effective catalysts in promoting fluidization in these networks, other, potentially more effective catalytic reagents were explored to affect the thiol-thioester exchange with the goal of offering higher exchange rates, reduced loading, enhanced stability and other beneficial attributes. As these networks were formed in a few seconds at ambient temperature via a thiol-ene photopolymerization, the assessment of various catalysts and catalyst loadings on the exchange rate was readily and quickly accomplished. While holding the concentration of each catalyst the same (3.0 mol%, ~0.6 wt%), basic catalysts with varying pKa values (5.3–13.6, of the conjugate acids in water) were evaluated by comparing their normalized stress relaxation rates (constant applied strain of 10%, 90 min, RT, Supplementary Figure 16, overlaid stress relaxation graph of 4-tert-butylpyridine, DIPEA, DBU, and TMG). More basic amines, such as DBU (1,8-Diazabicyclo(5.4.0)undec-7-ene, pKa = 12.0) or TMG (1,1,3,3-Tetramethylguanidine, pKa = 14.0), were found to relax stress more rapidly than less basic amines, such as 4-tert-butylpyridine (pKa = 5.5) or Hunig's base (N,N-Diisopropylethylamine, DIPEA, pKa = 10.8). Consistent with the pKa of the free thiol in the network being ~10.4 (based on the pKa of 3-methyl mercaptopropionate in water), only amines with pKa's (conjugate acid in water) above this threshold exhibited significant stress relaxation. Density functional theory calculations corroborate the role of the base in accelerating the thiol-thioester exchange. In the absence of a base, the thiol-thioester exchange requires a high activation enthalpy (ΔH$^‡$) of ~34 kcal/mol (Supplementary Figure 8, tabulated computational results) and is not active at room temperature; no stress relaxation was observed under these conditions. In contrast, in the presence of a base capable of deprotonating the thiol, exchange between thiolate and thioester is facile with a significantly reduced predicted barrier (ΔH$^‡$) of ~4 kcal/mol, corroborating the observed rapid base-catalyzed thiol-thioester exchange at room temperature that enables thermoplastic behavior.

**Photoswitching from solid to fluid.** As fluidic behavior of the thioester networks was shown to be hinged upon the incorporation of all three components required for the thiol-thioester exchange (free thiol, thioester, and base), it was hypothesized, as stated above, that a solid to fluid phase transition could occur if a base catalyst was created in situ. We anticipated that a two-stage irradiation scheme could be a feasible approach to this phase transition, whereby the network is first formed via a photoinitiated thiol-ene polymerization with a photoinitiator activated in the visible spectrum (455 nm or higher, Supplementary Figure 17.)[24] which leaves a catalytic amount of a photodeprotectable base, which is subsequently released by UV or near-UV irradiation (405 nm or lower, Supplementary Figure 17),[20] distributed homogenously throughout the sample (Fig. 2a, top right). Irradiation of the resulting material at a lower wavelength (365 nm) releases base only in the exposed areas and spatially promotes the thiol-thioester exchange, effectively switching the phase of the material from solid to fluid in a bistable manner (Fig. 2a, bottom right). To this end, combination of the thioester diene monomer **TE2** (1 equiv), PETMP (1 equiv), a UV-deprotectable base (**PB**, 5 mol%), and a visible photoinitiator (**HABI-Cl**, 1 mol%)[24] produced a stable, low viscosity resin. Irradiation of this resin with 455 nm light (30 mW/cm$^2$) induced rapid polymerization with the accompanying network formation (~1 min), which was observed to proceed to full conversion of the alkene species (Supplementary Figure 18, ATR of thin film following polymerization). Without further irradiation these networks acted as typical cross-linked solids, showing no crossover

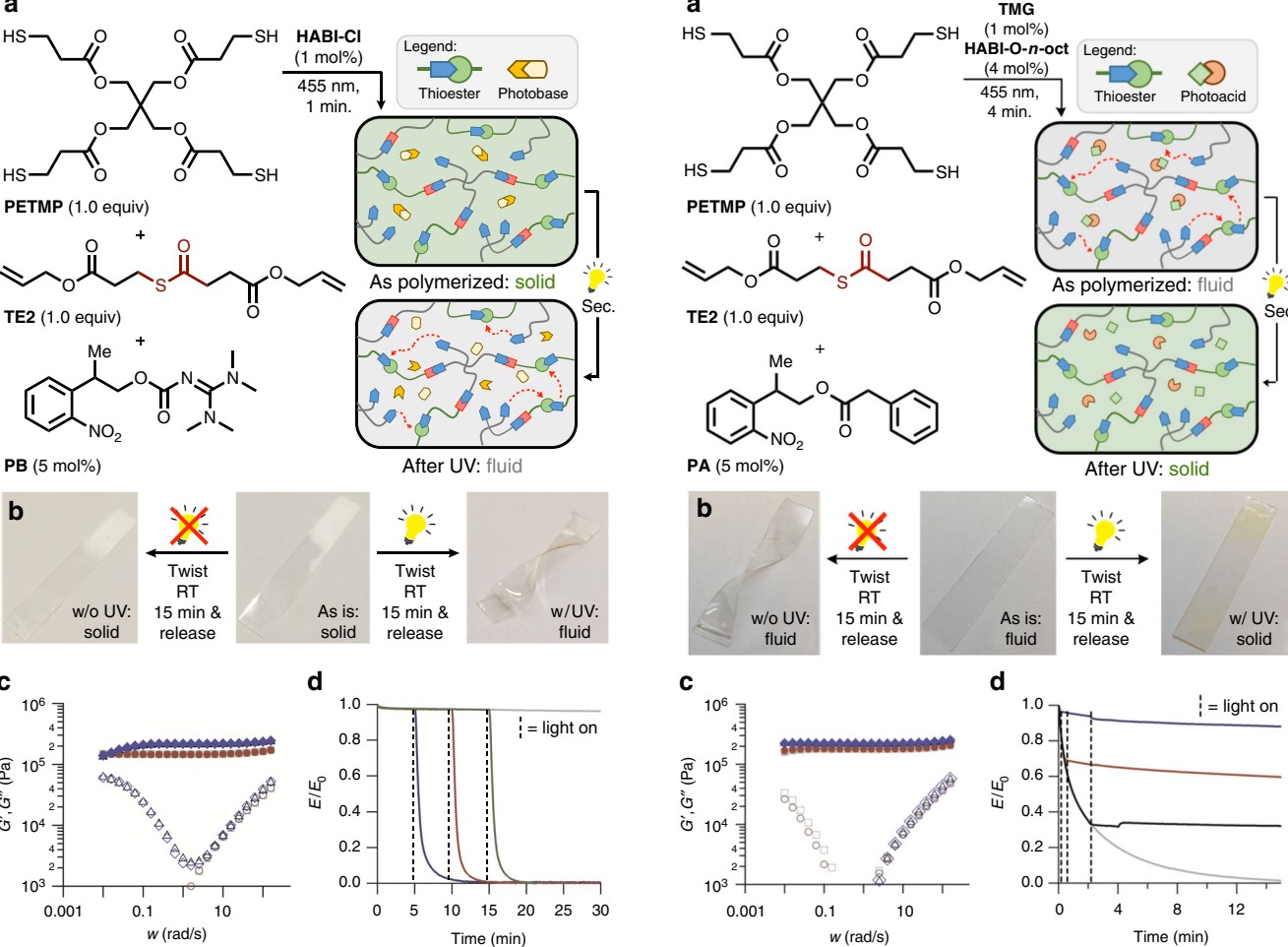

**Fig. 2** Permanent, instantaneous switching of a solid to a fluid. **a** A general formulation that gives a material which can be switched from a solid to a fluid with UV light. **b** As is, the solid sample does not adopt a new shape when twisted; however, after irradiation (5 min, 365 nm, 75 mW/cm²), the now fluid sample permanently adopted the twisted shape. **c** Rheometry shows that this material acts as a solid (immediately after polymerization: G′ = grey filled square; G″ = grey open square; 1.5 h after polymerization: G ′ = red filled circle; G″ = red open circle) until exposure to UV light, which switches it to a fluid (immediately after UV light: G′ = black filled triangle; G ″ = black open triangle; 1.5 h after UV light: G′ = blue filled diamond; G″ = blue open diamond). **d** The nearly instantaneous fluidization of the network upon exposure to UV light, shown here by the relaxation of stress at a constant strain (10% strain, light on at 5 (blue), 10 (red), and 15 (green) minutes, continuously irradiated, 365 nm, 75 mW/cm²). Grey line —not irradiated

**Fig. 3** Permanent, instantaneous switching of a fluid to a solid. **a** A general formulation that gives a material which is switched from a fluid to a solid with UV light. **b** As is, the fluidic sample forms a new permanent shape when twisted; however, after irradiation (5 min, 365 nm, 75 mW/cm²) the now solid sample does not adopt the new shape. **c** Rheometry shows that this material acts permanently as a fluid (immediately after polymerization: G′ = grey filled square; G″ = grey open square; 1.5 h after polymerization: G ′ = red filled circle; G″ = red open circle) until exposure to UV light, which switches it to a solid (immediately after irradiation: G′ = black filled triangle; G″ = black open triangle; 1.5 h after irradiation: G′ = blue filled diamond; G″ = blue open diamond). **d** The nearly instantaneous solidification of the material upon exposure to UV light, shown here by the relaxation of stress at a constant strain (10% strain, light on at 5 (blue), 20 (red), and 120 (black) seconds, irradiated for 120 s, 320–500 nm, 75 mW/cm², a small thermal recovery was noted in each case after the light was turned off). Grey line—not irradiated

of the storage (G′) and loss (G″) modulus at lower frequencies. However, upon further irradiation (365 nm, ~40 mW/cm², 10 min) the base catalyst (TMG) was released, catalyzing the thiol-thioester exchange in the network, and resulting in a network which acted as a viscous fluid with an increase in the loss modulus at lower frequencies (~1 rad/s, Fig. 2c). This solid to fluid switch was shown to be a permanent by re-testing the sample 1.5 h after the light was turned off, showing overlapping behavior of the two runs following the short irradiation. Traditionally, when solid thermosetting materials are strained, stress can build up, whereas fluids flow, rearrange and alleviate stresses. To this end, our initially solid material was placed under a constant strain (10%) which showed no stress relaxation, mirroring

the behavior of a solid. However, upon irradiation (365 nm, ~75 mW/cm², 5 min after the strain was applied, Fig. 2d) the basic catalyst was rapidly released, catalyzing the thiol-thioester exchange and rapidly relaxing all stress in the network. Further evidence of temporal control over solid to fluid phase transition was demonstrated by releasing the base 10 and 15 min after the strain was applied (Fig. 2d), in all three cases essentially all stress was relaxed within ~5 min once the material was stimulated by light.

**Photoswitching from fluid to solid.** In complement to the successful demonstration of a solid to fluid phase transition, the

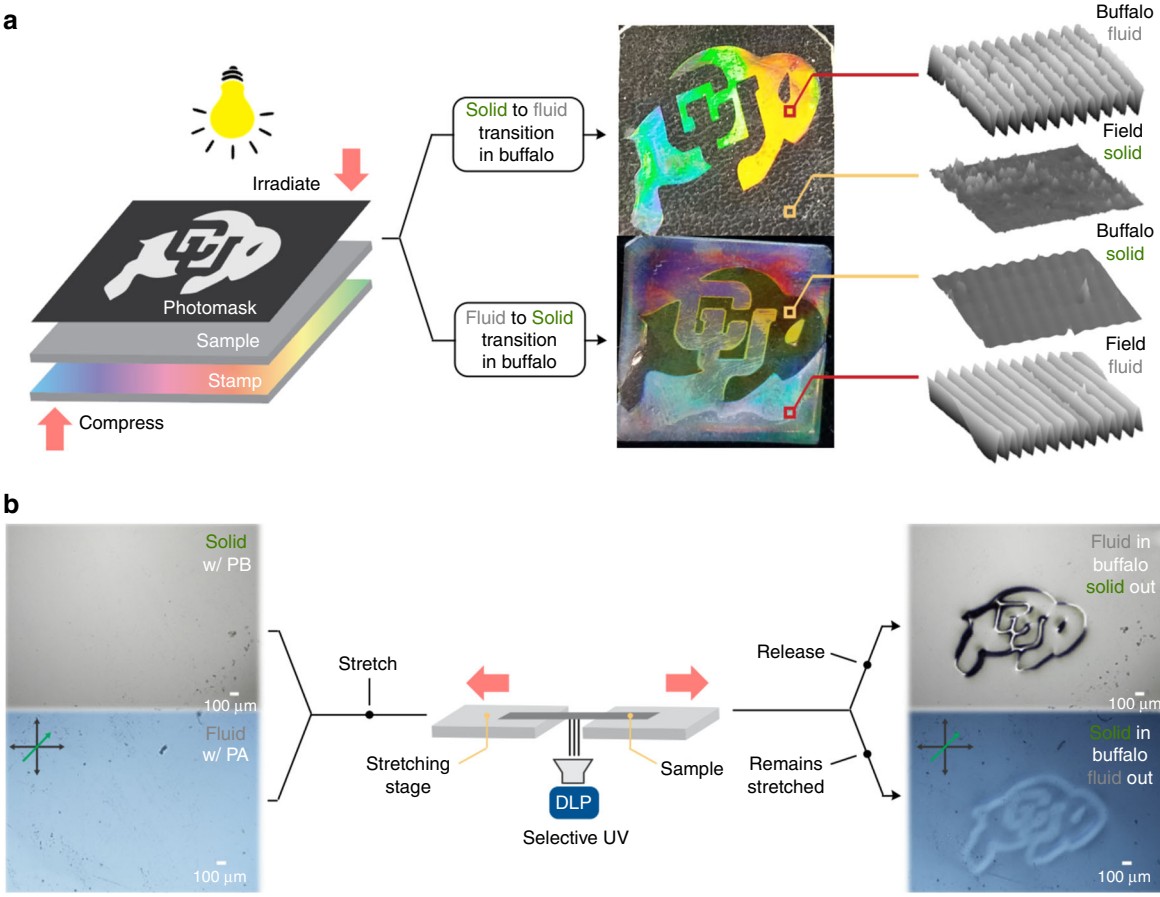

**Fig. 4** Macroscopic and microscopic spatial control over plasticity. **a** Spatial control over phase at a macroscopic scale. A nano-scaled pattern (noted by the multi-colored sections) was transferred from a stamp to the sample via compression; exposure to light dictated whether the sample responded as a fluid (top) or as a solid (bottom) in the unmasked, irradiated areas. **b** Spatial control over phase at a microscopic scale. Stretching under a microscope and irradiation with a DLP dictated whether the sample responded as a fluid (top) or as a solid (*bottom*) in selected, irradiated areas. Top images for the fluidizing formulation are brightfield microscope images (both images in a relaxed state) while the bottom images for the solidifying formulation are visualized through cross polarizers (both images in a stretched state)

development of a complementary switch (fluid to solid) would certainly be useful to practitioners who desire fluidic properties only initially during or immediately following the polymerization, in order to relax, for example, polymerization-induced stress or to set the material into a shape not otherwise feasible utilizing traditional fabrication techniques. To accomplish this switch, a base previously distributed in the network was neutralized by the photo-induced release of acid. As the NPPOC-protecting group was seemingly not affected by network formation via visible light (455 nm), but was highly sensitive to UV light (405 nm or lower), this approach was used (Supplementary Figure 19, overlaid UV-vis of **HABI-O-*n*-oct** and **PA**). Thus, an acid generator prepared by the condensation of phenylacetic acid and NPPOC-OH[21] was synthesized, which exhibited rapid kinetics for the acid release in solution (Supplementary Figure 20, NMR kinetics of acid release). Accordingly, the thioester diene monomer **TE2** (1 equiv), PETMP (1 equiv), a UV-deprotectable acid (**PA**, 5 mol%), a base catalyst (TMG, 1 mol%), and a visible photoinitiator (**HABI-O-*n*-oct**, 4 mol%)[24] were combined giving a stable, low viscosity resin which could be easily cast into a thin film or mold. Irradiation of this resin with 455 nm (50 mW/cm$^2$) light yielded formation of the network (8 min), which was noted to go to full conversion of the alkene species (Supplementary Figure 21, ATR of thin film following polymerization). Without further irradiation, this network acted as a viscous fluid, with an increasing loss modulus (G″)at lower frequencies (~1 rad/s, Fig. 3c). However, upon further

irradiation (320–500 nm, ~40 mW/cm$^2$, 10 min) the acidic catalyst was released, halting the thiol-thioester exchange in the network, and resulting in a network which acted as a solid (no crossover of the storage (G′) and loss (G″) modulus at lower frequencies). This fluid to solid switch was shown to be permanent by re-testing the sample 1.5 h after the light was turned off, showing overlapping behavior in the two runs following temporary irradiation. As stated above, fluids are capable of relaxing stresses once strained, however, solid materials are not. Consequently, our initially fluidic material was found to rapidly relax all stress in the network when placed under a constant strain (10%), mirroring the behavior of a fluid. However, upon irradiation (320–500 nm, 75 mW/cm$^2$, 5 s after strain was applied, Fig. 3d) the materials solidified, relaxing no additional stress. Further evidence of temporal control over fluid to solid phase transition was demonstrated by releasing the acid 20 and 120 s after the strain was applied (Fig. 3d). In all three cases no further stress was relaxed once the material was stimulated by light.

**Applications of photoswitchable states of matter.** Other than offering exquisite temporal control, photo-induced processes inherently allow for the contactless transfer of energy to substrates with spatial control. Utilizing these switchable materials, the balance between solid and fluid behavior is controlled on both the macroscopic and sub-micron scale. Nanoimprint Lithography

(NIL), a direct-contact lithographic technique, is capable of fabricating optically active surface topographies on polymeric materials.[25, 26] Here, NIL is used to highlight the spatial control intrinsic to both the solid to fluid and fluid to solid switches by compressing films into a uniform, optically diffractive mold and controlling which areas of the film surface reproduce the mold's topography due to their fluidic behavior. With both switchable materials, it was readily shown that transfer of the surface topology of the stamp to the material could be spatially controlled using light by providing (solid to fluid) or depriving (fluid to solid) the base catalyst in the shape of a buffalo (photomask: 2.5 cm × 2.5 cm, Fig. 4a). Verification of this spatial topology transfer was confirmed by atomic force microscopy (AFM, Fig. 4a, right). Mechanophotopatterning, or the coupling of mechanical strain and light to spatially induce changes in the material,[27] was utilized to demonstrate microscopic control (100 μM features) over plasticity. To this end, the samples were fixed to a stretching device under a microscope equipped with a dynamic light projector (DLP). Separately taking both types of switchable films and stretching while irradiating the outline of a buffalo, holding in the stretched state (developing), and releasing the strain resulted in deformation only inside of the buffalo (solid to fluid phase transition), as viewed in the bright field, or conformational changes only outside of the buffalo (fluid to solid phase transition), as viewed through cross polarizers (Fig. 4b).

**Discussion.** As evidenced above, coupling of the thiol-thioester exchange with photo-responsive reagents yielded materials that could transition between phases (fluid to solid and vice versa) on-demand with nearly perfect spatial and temporal control. This platform significantly increases the level of sophistication and command accessible to practitioners utilizing cross-linked polymer networks. Indeed, complex, multi-step molding schemes or on-demand recycling are easily envisioned using this system. Although the switches used above relied on destructive, fragmentation mechanisms to provide the active reagents, the use of non-destructive, photo-isomerizable reagents to switch many times between solid and fluid phases with spatial and temporal control is envisioned. The dynamic thioester cross-link developed here will no doubt find universal applicability in additional cross-linked network polymers to imbue them with plastic behavior with application towards low-stress, recyclable stereolithography/3D printing resins and diffraction-limited optical devices.[28]

## Methods

**Synthesis of a fluidic thioester network**. To a 10.0 mL speed mixer vial was added 250 mg (0.87 mmol, 1.00 equiv) of **TE2**, 427 mg (0.87 mmol, 1.00 equiv, 100% excess thiol) of pentaerythritol tetra(3-mercaptopropionate) (PETMP), and 36.4 μL (30.2 mg, 0.17 mmol, 0.20 equiv, 20.0 mol%) of N,N,N',N″,N″-pentamethyldiethylenetriamine (PMDETA) each via Pasteur pipettes. This clear resin was then manually mixed with a pipette tip for ~2 min to make a homogenous mixture. Following this, approximately 8.91 mg ($3.48 \times 10^{-2}$ mmol, 0.04 equiv, 4.00 mol%) of 2,2-dimethoxy-2-phenylacetophenone (DMPA), which had been crushed with the flat side of a spatula to form a fine powder, was added and the resin was further manually mixed with a pipette tip for an additional ~2 min to form a homogeneous mixture. At this time the clear resin was poured between two glass slides treated with Rain-X (ITW Global Brands, Houston, TX) using 250 μm thick spacers (Small Parts Inc., Logansport, IN). The material was irradiated (365 nm, 5.00 μW/cm$^2$, room temperature) for ~10 min to yield the thiol excess, thioester-containing network polymer. The conversion was found to be essentially quantitative by in situ IR, revealing complete consumption of the alkene species (Supplementary Figure 9). Moreover, it was shown that, due to the quantitative nature of the thiol-ene reaction, any excess of either reactant (ene or thiol) remained unreacted in the final network polymer (Supplementary Figure 10).

**Data availability**. All data are available on request from the authors.

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

## Acknowledgements

This work was supported by grants from NSF-MRSEC (DMR 1420736), the NSF (CHE-1214109 and DMR-1310528), DARPA/US Army (W911NF-14-1-0605), NSF Industry/University Cooperative Research Center for Fundamentals and Applications for

Photopolymerizations, the Arnold and Mable Beckman Foundation postdoctoral fellowship (B.T.W.), the U.S. Department of Education GAANN Fellowship in Functional Materials (G.B.L), and an NSF pre-doctoral graduate research fellowship (M.K.M). We thank Prof. Jeffrey Stansbury (CU Boulder), Prof. Robert McLeod (CU Boulder), Dr. Joe Oxman (3M), Dr. Wayne Mahoney (3M), Dr. Tai-Yeon Lee (DSM), Dr. Johan Jansen (DSM), Prof. Will Gutekunst (Georgia Tech), and Marty Stanton (Mosaic Biosciences) for helpful scientific discussions.

## Author contributions

B.T.W., M.K.M., G.B.L., L.M.C., C.W., S.M., C.H.-L., H.M.C., C.B.M., Y.D., and C.N.B. participated in the conceptual development of this research, data collection/analysis, and preparation of this manuscript.

## Additional information

**Competing interests:** The authors declare no competing interests.

