## [Peer Review File · Nature Communications]

Reviewers' comments:

Reviewer #1 (Remarks to the Author):

The authors report on the dynamic chemistry of thioesters in the presence of thiols and their elegant incorporation into a polymer material via a classical thiol-ene photopolymerization. Critically, the thiol-thioester exchange reaction is prone to base-promoted catalysis by which the material – that contains an excess of dangling thiol functionalities – is capable of switching from a solid phase behaviour (network) into a liquified one. Moreover, photocaged bases were used in order to implement light (instead of classical heat or pressure) as an external trigger to (spatially) control the exchange reaction and by extension create an on/off switch over the materials' state. The reverse phase transitions (i.e. liquid to solid) was affected via a similar approach, now using a photocaged acid that neutralizes the base catalyst and thereby suppresses the exchange reaction. The practicality of the designed light-switchable materials is demonstrated by an inspiring lithographic application.

Overall, the concept of switchable states of matter is presented in a convincing and quite innovative manner and sufficient information in support of the findings is provided. The applicability of the presented system, evidenced by mechanopatterning experiments, is extremely well-received and is believed to enable a step change in the future design of photoswitchable materials, ideally containing reversible photoswitchable linkages to toggle states of matters on and off.

I would therefore recommend publication in Nature Communications, after having addressed the following remarks:

1. The authors provide a lot of Supplementary information (35 pages), which no doubt is strongly encouraged, but do not refer to the appropriate sections within the manuscript text. Simply implementing a standardized cross-reference, i.e. "see Supplementary Information for details" is not sufficient to point the reader in the right direction. In fact, taking lines 100-109 as an example, this standard comment is repeated far too often and even becomes irritating to read. Instead, the authors should refer to the specific Supplementary Figure, Table or section, ideally followed with a few words to explain the reader what can be evidenced there.
2. On line 72-75, the authors mentioned that the exchange reaction was computed to have significantly lower kinetic barriers relative to the analogous transesterification. This rises confusion when looking at Supplementary Fig. S2 (page 16), where the calculated barriers for the deprotonated thiolate-thioester reaction is very similar – and in fact even slightly higher – than the methoxide-methylester reaction. Is the distinction in kinetic barriers than merely a result of the base catalyst used (i.e. $pK_a \leq 10$)? Change of words might be considered in order to avoid confusion.
3. Looking at the structure of monomer TE2 (and of the PETMP tetravalent thiol), quite an excess of ester relative to thioesters is present throughout the network. This might promote some kind of cross-exchange reaction of the thiol anion with the ester instead of the thioester functionalities. Is such an exchange reaction studied/detected (control experiment)? And to what extent (not)? Related to comment 4, I also do not see computational results of the cross-exchange reaction in Supplementary Fig. S2. This might bring more clarity.
4. A key control experiment, evidencing the photostability of the thioester monomer under the applied conditions of irradiation, is missing.
5. Is there an explanation for the slower stress relaxation in Figure 2d when the light is switched on after a dead time of 5 minutes compared to 10 and 15 minutes? The former seems to have a longer tailing compared to the other two measurements.
6. The authors provide the synthesis of a diallylester control monomer (DAEC) in the supplementary information but do not mention this compound, nor its use, in the manuscript text. Why has this compound been used and to what extent?
7. This is not the first time that the authors highlight this thioester exchange and thus most probably they addressed the stability of free thiols (prone to oxidation) in the crosslinked networks

in a previous publication. There should be a reference to this (if not, discuss it in more detail). A nice control experiment to demonstrate the stability of the solid state could be to check G' and G'' after few weeks (or months). While paper shows proof of concept, implementation is only valid if longer term stability can be guaranteed.

Additional minor points:

1. After several iterations, a scalable synthesis of a suitable thioester monomer, that meets certain requirements, is achieved. Considering the synthesized compound TE1 is seen as a "core monomer", I would like to see the structure of TE1 included within the manuscript file (could be incorporated in first figure).
2. Numerous abbreviations, both in the manuscript file as well as in the Supplementary Information, are not defined at first use. This makes it slightly more challenging towards a full understanding of the work.
3. Line 33-36: please provide an appropriate reference when using "in analogy to ...".
4. The caption of Figure 1 should be revised. A description for only 4 out of 5 panels is provided.
5. Line 55, cross-reference (Fig. 1a, left): both the transition of solid to liquid and of liquid to solid might be interpreted to occur both at elevated temperature when discussing the phase transition behaviour of vitrimers. It would be better to replace the two symbols by "cooling" and "heating", or by "T+" and "T-".
6. Line 106-107: it should be briefly mentioned how the PMDETA base is covalently attached to the network.
7. Line 123-124: providing pKa values of DBU, TMG and Hünig's base within the text would allow for a better understanding.
8. Line 242: adjust "100?µM"
9. Supplementary Information: more white spacing is needed between the end of a synthesis procedure and the following reaction scheme, to provide a better overview.
10. Figure S1: the bottom structure is represented in an ambiguous manner and could be interpreted as the addition product with trimethyl amine instead of any nucleophilic attack.
11. Caption Figure S2: please include the basis set used for the computations.
12. Please change mgs, mLs, grams, etc... to mg, mL, gram, etc... throughout the entire manuscript and Supplementary information file.

Reviewer #2 (Remarks to the Author):

In their manuscript Bowman and co-workers report the design, synthesis and characterisation of a covalent adaptable network that can undergo a switch from solid to fluid and vice versa, presenting thus a stable biphasic system. Overall the design of the polymer material is convincingly demonstrated with clear rationale, while the experimental evidence is also comprehensive and compelling. Moreover, the topic, covalent polymer networks is a timely subject of clear interest to the readership of Nature Communications. Assuming the authors can address my few remaining questions, I suggest acceptance of this manuscript for Nature Communications.

My main point is related to the overall description of the material that the authors report on. In their words: "[...] Here, we demonstrate a responsive material which switches phases, solid to fluid and vice versa, at room temperature upon exposure to light; this phase change is bistable, [...]" (lines 24-25). This phrasing seems to suggest that one single material can undergo both transitions (solid to liquid AND liquid to solid), however the authors only show a material that EITHER can undergo a solid-to-liquid transition OR a liquid-to-solid transition. I recommend the authors to address this issue, either by commenting on the (im)possibility of making a truly repeatably switching material, or by rephrasing their material description.

Another question I have is related to the thermal stability of the photoresponsive base (PB) and acid (PA). Are both molecules stable in the dark? Also, how fast/slow do they generate the base /

acid under ambient light? Depending on the rate of this 'background' decomposition, the claimed stability of the material under non-illumination conditions needs to be commented. That is, if the thermal composition is rather fast (I note that the time scale figure 3d is seconds, which is a fairly short time scale, leaving the question about the thermal stability on the time scale of minutes / hours unaddressed) the material does not start out in a genuinely stable state, but in a metastable state.

In the caption of figure 1 the overall title refers simply to 'matter'. With respect to the application of temperature as a stimulus, this leads to confusion: as the authors explain in lines 51-55, in covalent adaptable networks temperature can act as a kinetically controlled stimulus. However, for matter 'in general' temperature changes the thermodynamic stability of matter: i.e. lowering water from +5 °C to -5 °C changes the thermodynamically most stable state from liquid to solid. To avoid this confusion, I would suggest to specify in the caption that the figures a and b refer to covalent adaptable networks.

Some minor points:

- * The caption of figure 1 is incomplete: panel c of the figure is not described in the caption.
- * Line 73: in what sense is the exchange process 'robust'?
- * In scheme on page S8 and S9 of the SI, the shown diallyl thioester is incorrectly labelled as TE1; it should be labelled TE2.

Response to reviewer 1:

Major revisions

- 1) “The authors provide a lot of Supplementary information (35 pages), which no doubt is strongly encouraged, but do not refer to the appropriate sections within the manuscript text. Simply implementing a standardized cross-reference, i.e. “see Supplementary Information for details” is not sufficient to point the reader in the right direction. In fact, taking lines 100-109 as an example, this standard comment is repeated far too often and even becomes irritating to read. Instead, the authors should refer to the specific Supplementary Figure, Table or section, ideally followed with a few words to explain the reader what can be evidenced there.”

Response: More specific references to supplementary figures, tables, and sections have been added to the text to assist in further guiding the reader.

- 2) “On line 72-75, the authors mentioned that the exchange reaction was computed to have significantly lower kinetic barriers relative to the analogous transesterification. This rises confusion when looking at Supplementary Fig. S2 (page 16), where the calculated barriers for the deprotonated thiolate-thioester reaction is very similar – and in fact even slightly higher – than the methoxide-methylester reaction. Is the distinction in kinetic barriers than merely a result of the base catalyst used (i.e. $pK_a \leq 10$)? Change of words might be considered in order to avoid confusion.”

Response: We removed the text in the paper and made specific note in the supporting information regarding that the observed activity of the thiol-thioester exchange is likely due to the ability of free thiols to form thiolate species with the mild bases employed.

- 3) “Looking at the structure of monomer TE2 (and of the PETMP tetravalent thiol), quite an excess of ester relative to thioesters is present throughout the network. This might promote some kind of cross-exchange reaction of the thiol anion with the ester instead of the thioester functionalities. Is such an exchange reaction studied/detected (control experiment)? And to what extent (not)? Related to comment 4, I also do not see computational results of the cross-exchange reaction in Supplementary Fig. S2. This might bring more clarity.”

Response: We have not observed any exchange in the control (without thioester) as was illustrated in **Figure S5**. The difference between the control and the exchanging system was merely a singular thioester which was replaced by two methylenes (**DAEC**).

- 4) “A key control experiment, evidencing the photostability of the thioester monomer under the applied conditions of irradiation, is missing.”

Response: This control would presume that thiyl radicals could participate in the exchange or thioesters were activated by irradiation, which we have shown based on the very high kinetics of the thiol-ene reaction (Figure **S3**) that this does not occur. Indeed, the presence of the thioester in stoichiometric quantities has no effect on the rate of this reaction. If the thioester were acting as a RAFT reagent or interacting with the thiyl radical concentration, the rate would slow precipitously. Indeed, it has been shown that the thioester functional group is orthogonal to radical induced polymerizations/processes (see: Neindre, M. L., Magny, B., Nicolaÿ, R. Evaluation of thiocarbonyl and thioester moieties as thiol protecting groups for controlled radical polymerization *Polym. Chem.*, **4**, 5577 (2013)). Moreover, if some concentration of thiolate were formed during irradiation, again, the rate of polymerization would be effected (see: Love, D. M., Kim, K., Goodrich, J. T., Fairbanks, B. D., Worrell, B. T., Stoykovich, M. P., Musgrave, C. B., Bowman, C. N. Amine induced retardation of the radical-mediated thiol-ene reaction via the formation of metastable disulfide radical anions *J. Org. Chem.*, **83**, 2912 (2018)). Based on the depth of literature which shows the lack of participation of the thioester in light induced processes, we believe that showing this control is not necessary.

- 5) “Is there an explanation for the slower stress relaxation in Figure 2d when the light is switched on after a dead time of 5 minutes compared to 10 and 15 minutes? The former seems to have a longer tailing compared to the other two measurements.”

Response: We believe that this difference is within experimental error and could be explained by a small change in the intensity of the light or error in the measurement of sample dimensions. Regardless of the difference, the trend is consistent across all of the samples.

- 6) “The authors provide the synthesis of a diallylester control monomer (DAEC) in the supplementary information but do not mention this compound, nor its use, in the manuscript text. Why has this compound been used and to what extent?”

Response: This molecule (**DAEC**) was utilized as the “– thioester” control in **Figure S5**. A specific note was made above the synthetic procedure in the supporting information to avoid confusion.

- 7) “This is not the first time that the authors highlight this thioester exchange and thus most probably they addressed the stability of free thiols (prone to oxidation) in the crosslinked networks in a previous publication. There should be a reference to this (if not, discuss it in more detail). A nice control experiment to demonstrate the stability of the solid state could be to check G' and G'' after few weeks (or months). While paper shows proof of concept, implementation is only valid if longer term stability can be guaranteed.”

Response: The paper in question has been added as a citation (28).

Minor revisions

All requested minor changes have been made to the text and supporting information.

Response to reviewer 2:

Major revisions

- 1) “My main point is related to the overall description of the material that the authors report on. In their words: “[...] Here, we demonstrate a responsive material which switches phases, solid to fluid and vice versa, at room temperature upon exposure to light; this phase change is bistable, [...]” (lines 24-25). This phrasing seems to suggest that one single material can undergo both transitions (solid to liquid AND liquid to solid), however the authors only show a material that EITHER can undergo a solid-to-liquid transition OR a liquid-to-solid transition.”

Response: Lines 24-25 have been updated to reflect that two materials and not a singular material were employed to show switching.

- 2) “Another question I have is related to the thermal stability of the photoresponsive base (PB) and acid (PA). Are both molecules stable in the dark? Also, how fast/slow do they generate the base / acid under ambient light? Depending on the rate of this ‘background’ decomposition, the claimed stability of the material under non-illumination conditions needs to be commented. That is, if the thermal composition is rather fast (I note that the time scale figure 3d is seconds, which is a fairly short time scale, leaving the question about the thermal stability on the time scale of minutes / hours unaddressed) the material does not start out in a genuinely stable state, but in a metastable state.”

Response: Both molecules are quite stable in the dark, as has been reported previously by us and others previously (see citations 20 and 21 in the manuscript for further information regarding this). Indeed, the photoresponsive acid and base samples utilized in this manuscript were each, according to lab notebooks, 1-1.5 years old at the time they were employed in our formulations. Moreover, the UV-vis spectra of these molecules are provided in the supporting information (Figure **S11** and **S13**); these graphs show that photoresponsive molecules based on an NPPOC core only trail into the visible spectrum (405 nm), thus require a large dose of UV light to initiate decomposition. We believe that they do indeed start out in a genuinely stable state and are only activated when irradiated with UV light (365 nm).

In the caption of figure 1 the overall title refers simply to ‘matter’. With respect to the application of temperature as a stimulus, this leads to confusion: as the authors explain in lines 51-55, in covalent adaptable networks temperature can act as a kinetically controlled stimulus. However, for matter ‘in general’ temperature changes the thermodynamic stability of matter: i.e. lowering water from +5°C to –

5°C changes the thermodynamically most stable state from liquid to solid. To avoid this confusion, I would suggest to specify in the caption that the figures a and b refer to covalent adaptable networks.

3) We've made this change to the manuscript.

Minor revisions

All requested minor changes have been made to the text and supporting information.

REVIEWERS' COMMENTS:

Reviewer #1 (Remarks to the Author):

Referee Task for Nature Communications:
"Bistable, Photoswitchable States of Matter".

The authors have revised their manuscript and addressed the critical points risen. The re-write has cleared out some unambiguous understandings, yet some (minor) concerns have been left answered unsatisfactorily.

I still find it critical, for transparency, that the "- thioester" control experiment with DAEC is mentioned in the main text (cf. line 105) and not solely in the supplementary information file. Furthermore, a few requested minor changes were not addressed. I still do not see the structure of the scalable core monomer "TE1" incorporated in the main text file, which I consider important for the broad audience of Nature Communications to understand. A reference related to "in analogy to mechanically alterable switches used to operate electrical lighting displays" is also still missing. Given that the above stated (minor) points will be addressed, I suggest this work is ready for publication in Nature Communication.

Reviewer #2 (Remarks to the Author):

On the basis of their revised manuscript and rebuttal that satisfactorily address the points raised by the reviewers, I now fully recommend acceptance of the manuscript.

A final suggestion from my side is to include in their manuscript (SI would suffice, with a reference in the main text) the response on question 4 by reviewer 1. I consider it a valuable and insightful discussion for the reader.

Response to reviewers:

Reviewer #1:

- 1) I still find it critical, for transparency, that the “- thioester” control experiment with DAEC is mentioned in the main text (cf. line 105) and not solely in the supplementary information file.

Response: We have updated the manuscript to include this critical point.

- 2) I still do not see the structure of the scalable core monomer “TE1” incorporated in the main text file, which I consider important for the broad audience of Nature Communications to understand.

Response: We have changed Figure 1 to include the structure of the scalable core monomer “TE1”.

- 3) A reference related to “in analogy to mechanically alterable switches used to operate electrical lighting displays” is also still missing.

Response: We have removed “in analogy to mechanically alterable switches used to operate electrical lighting displays” from the text of the manuscript.

Reviewer #2:

- 1) A final suggestion from my side is to include in their manuscript (SI would suffice, with a reference in the main text) the response on question 4 by reviewer 1. I consider it a valuable and insightful discussion for the reader.

Response: We have added a note regarding this in the Supplementary Discussion of the Supplementary Information file.